# Identification of the Key Pathways and Genes Involved in the Wax Biosynthesis of the Chinese White Wax Scale Insect (*Ericerus pela* Chavannes) by Integrated Weighted Gene Coexpression Network Analysis

**DOI:** 10.3390/genes13081364

**Published:** 2022-07-29

**Authors:** Wei-Feng Ding, Xiao-Fei Ling, Qin Lu, Wei-Wei Wang, Xin Zhang, Ying Feng, Xiao-Ming Chen, Hang Chen

**Affiliations:** 1Institute of Highland Forest Science, Chinese Academy of Forestry, Kunming 650224, China; dingwf@caf.ac.cn (W.-F.D.); lingxf0305@foxmail.com (X.-F.L.); lqin@caf.ac.cn (Q.L.); sk121www@caf.ac.cn (W.-W.W.); zhangxincindy126@126.com (X.Z.); 2Key Laboratory of Breeding and Utilization of Resource Insects, National Forestry and Grassland Administration, Kunming 650224, China; rirify@139.com (Y.F.); cafcxm@139.com (X.-M.C.)

**Keywords:** weighted gene coexpression network analysis (WGCNA), transcriptome sequencing, gene network, fatty acid metabolism, wax biosynthesis

## Abstract

The white wax secreted by the male insects of the Chinese white wax scale (CWWS) is a natural high-molecular-weight compound with important economic value. However, its regulatory mechanism of wax biosynthesis is still unclear. In this study, a weighted gene coexpression network analysis (WGCNA) was used to analyze transcriptome data of first- and second-instar females, early and late female adults, and first- and second-instar males. A total of 19 partitioned modules with different topological overlaps were obtained, and three modules were identified as highly significant for wax secretion (*p* < 0.05). A total of 30 hub genes were obtained through screening, among which elongation of very-long-chain fatty acids protein (ELOVL) and fatty acyl-CoA reductase (FAR) are important catalytic enzymes of fatty acid metabolism. Furthermore, their metabolic catalytic products are involved in the synthesis of wax biosynthesis. The results demonstrate that WGCNA can be used for insect transcriptome analysis and effectively screen out the key genes related to wax biosynthesis.

## 1. Introduction

The CWWS, *Ericerus pela* Chavannes (Hemipetera: Ericerus) is a resource insect with important economic value [1]. The white wax secreted by the second-instar male of CWWS is a natural high-molecular-weight compound consisting largely of ceryl cerotate (C_25_H_51_COOC_26_H_53_) together with other wax esters [2], which is widely used in machinery, food, the chemical industry, and other industries [3]. The WWS displays dramatic sexual dimorphism, with notably different metamorphic fates in males and females [4]. The females develop through three stages: egg, larva (two nymphal instars), and adult. They live fixed parasitic lives on the host tree, and their body walls develop into highly keratinized chitin shells that play a role in defending against natural enemies and pathogens, maintaining water in the body, and protecting the insect from sunburn. The males develop through the stages of egg, larva (two nymphal instars), pupa (prepupa and pupae), and adult. The male insect body wall is weakly keratinized. At the first instar of the male larva, less wax silk is secreted, and at the second instar, a large amount of wax silk is secreted to form a wax-covered insect body that plays a role in defending against natural enemies, isolating the external adverse climate, and maintaining the normal physiological metabolism of the insect body [5].

In the production of white wax, farmers have formed a production model of “producing insects in high mountains and producing wax in low mountains” after nearly a thousand years of cultivation of CWWS. In the Yunnan Guizhou Plateau and other mountainous areas of China, the female of the CWWS is harvested and transported to Sichuan, Hunan, and other places for the production of white wax. The influence of local adverse environmental factors on the CWWS is used to improve the wax excretion. For example, the high temperature and humid environment makes the males secrete too much white wax to protect themselves. However, this production mode requires the long-distance transportation of species from insect-producing areas such as Zhaotong and Yunnan to wax-producing areas [6]. In this process, there are disadvantages, such as a large loss of seed insects and high production costs. Therefore, revealing the wax secretion mechanism of CWWS has significance for wax production [7].

The study of wax ester biosynthesis was first carried out in *Arabidopsis thaliana* [8]. With the development of genetic and biochemical analysis techniques, such as the isolation and identification of cuticle wax deletion mutants, gas chromatography/mass spectrometry, and isotope tracing, the wax biosynthesis pathway of the plant cuticle was studied. The FAR, wax synthase (WS), and β-ketoacyl-CoA synthase (KCS) were verified as the key enzymes [9,10,11,12,13]. It was found that the main component of the sebum in the prepuce gland of mice was wax monoester [14,15]. To prove whether the synthesis route of mammalian wax esters is similar to that of plant oils, they expressed the *FAR* and *WS* genes of mice in HEK293 cells and used palmitic acid (hexadecanoic acid) as a substrate for the catalytic reaction to obtain fatty alcohols and wax esters, thus confirming that the synthesis of wax monoesters in mammals involves a two-step biosynthesis pathway catalyzed by FAR and wax synthase enzymes. The synthesis process of insect wax esters is also very similar. Long-chain fatty acids are reduced to corresponding fatty alcohols under the catalysis of FAR, and long-chain fatty acids and fatty alcohols are formed into esters under the action of WS. These studies show that the wax ester synthesis pathway is conserved in plants and animals.

Research on the wax biosynthesis of the CWWS insects began in 2012. Referring to previous research methods on wax ester metabolism in animals and plants, Yang conducted high-throughput sequencing of the transcriptome of the second-instar males at the peak of the white wax secretion [16]. Through sequence annotation, the sequences of the predicted *FAR* and *WS* genes of CWWS were obtained. In the following years, Yang’s team constructed the cDNA library of CWWS, cloned the complete sequences of *FAR* and *WS*, and used the insect cell–baculovirus expression vector system to obtain the final protein [17,18,19]. The C_28_ fatty acyl-CoA was used as the reaction substrate and successfully reduced to C_28_ fatty alcohol. It was confirmed that FAR was involved in the synthesis of white wax. By analyzing the transcriptome data of the CWWS of different insect stages and sexes, Liu et al. [3,20] found that the exogenous juvenile hormone analogue (JHA) and ecdysone (20E) hydroxylase (CYP18A1) that regulate the development of the CWWS insect bodies and insect states have regular differential expression in different insects with the process of metamorphosis. In the validation experiment, spraying a certain concentration of JHA on the second-instar male larvae of the CWWS significantly increased the wax secretion of the larvae. Although these results provide a reference for understanding the biochemical mechanism of wax secretion, the wax ester regulation pathway of the CWWS insect is still not clear.

With the continuous advancement of sequencing technology and the gradual reduction in sequencing costs, more and more researchers have begun to design multi-sample RNA-seq studies. Multiple samples can analyze gene expression changes under multiple conditions, making scientific research data more substantial, systematic, and convincing. However, at the same time, multi-samples bring a large amount of grouped data, and the workload required for the traditional difference analysis between two groups is very large, which is not conducive to efficient research. WGCNA can summarize and organize complex data and efficiently study the overall expression rules of genes. At the same time, it can systematically obtain the interaction pattern between genes in the samples, help to mine key genes, predict gene function, and achieve the purpose of significantly improving gene screening. Therefore, WGCNA plays an important role in analyzing multi-sample RNA-seq data. We obtained the transcriptome data of the CWWS samples of different sexes, insect statuses, and ages as well as the corresponding wax secretion values of each test insect. We used the WGCNA [21] method to analyze the data in this study to obtain the hub genes associated with CWWS wax secretion.

## 2. Materials and Methods

### 2.1. Sample Collection

The eggs of the CWWS insects originated in Zhaotong, Sichuan, China (northern latitude: 27°33′, east longitude: 103°72′, altitude: 1685 m). The samples of the CWWS insects were collected in the Institute of Highland Forest Science, Chinese Academy of Forestry (northern latitude: 25°3′20″, east longitude: 102°45′16″, altitude: 1950 m). The parasitic tree was *Ligustrum lucidum*. Moreover, first- and second-instar females (*FF* and *SF*), early and late female adults (*EA* and *LA*), and first- and second-instar males (*FM* and *SM*) were collected with three biological replicates for each sample. All 18 samples were collected and stored frozen at −80 °C.

### 2.2. High-Throughput Transcriptome Sequencing

The methods of total RNA extraction, transcriptome data assembly and annotation, and differential gene analysis and enrichment have been published by Liu [20] and will not be repeated in this article.

### 2.3. Weighted Correlation Network Analysis

Referring to the tutorial written by Peter Langfelder and Steve Horvath [22], we completed the network analysis of the CWWS expression data and obtained modules related to wax secretion. We used the R language (version 4.2.0, the R foundation) package WGCNA (version 1.71, Peter Langfelder and Steve Horvath, Los Angeles, CA, USA) [21] as the analysis software, using RStudio (version 2022.02.3Build492, RStudio Public Benefit Corporation, Boston, MA, USA) as a tool for code writing and execution. The hardware platform running the code was a DELL PowerEdge T630 server with dual Intel Xeon CPU E5-2697 v4 processors and 512 GB of memory. The operating system was Ubuntu 22.04 LTS. The analysis process consisted of the following four main steps.

#### 2.3.1. Data Input and Preprocessing

There was a total of three data tables for WGCNA analysis (Appendix A), which were the fragments per kilobase of exon per million (FPKM) value list of all samples, the wax secretion trait table of each sample, and the gene annotation table. First, the FPKM value list was detected, and the genes whose expression levels were 0 in all samples were eliminated. Then, we calculated the median absolute deviation (MAD) for each gene and sorted by this value. We retained genes with the top 75% of MAD that were greater than 0.01. Unqualified gene data were removed, and the obtained data table was used for a cluster analysis of the samples. We checked whether there were any obvious outliers by looking at the obtained sample clustering tree. If abnormal samples were found, this set of sample data was removed.

#### 2.3.2. Gene Network Construction and Module Identification

First, we used the *pickSoftThreshold()* function in the WGCNA software package to analyze the network topology for various soft-thresholding powers and selected the appropriate soft-thresholding power (*β*) through the scale-free fit index. The block-wise network construction and module detection were performed using the *blockwiseModules()* function with the *β* as a parameter. This step was the most time-consuming of all analyses.

#### 2.3.3. Relating Modules with Wax Secretion Trait Data

The *cor()* function was used to calculate the correlation between the module and the trait data (the information table of wax secretion of each sample). Then, the *corPvalueStudent()* function was used to calculate the Student asymptotic *p*-value of the correlation and draw the module–trait associations heatmap. We selected modules with high weight correlations (modules with *p*-values < 0.05) to draw a scatter plot of gene significance and module membership.

#### 2.3.4. Network Visualization

We used the *exportNetworkToCytoscape()* function to filter out the modules with *p*-values < 0.05 from the topological overlap results and generate the edge and node data files that can be imported by the Cytoscape (version 3.9.1, Cytoscape Consortium) software [23].

We have provided the R language source code of the entire calculation process described above with detailed annotations; see Appendix A.

### 2.4. Selection of Hub Genes

Due to the excessive number of genes contained in some modules, the generated edge file was too large, and the calculation of the node degree using Cytoscape was very slow. Therefore, we first preprocessed the edge file. Gene nodes not annotated to the Swiss-Prot protein database were first removed from the network. Then, we calculated the degree of each node, that is, how many edges each node had, and sorted them from large to small. We obtained the list of the top 100 nodes (genes) with the maximum degree, then loaded the module with Cytoscape and created a new network by selecting these 100 nodes and all the edges connecting these nodes. In this new network, the degrees of these 100 nodes were calculated using CytoNCA (version 2.1.6, Yu Tang et al., Changsha, China) [24], a plugin for Cytoscape. We then had to check the “with weight” option when calculating and setting the weight parameter in the “Edges attributes” panel. From the analysis results of CytoNCA, the top 10% of proteins, ranging from large to small, were selected as hub genes.

### 2.5. Drawing of Figures

The toning and layout of all figures (Figure 1, Figure 2, Figure 3, Figure 4, Figure 5, Figure 6, Figure 7 and Figure 8) were completed using authorized Microsoft PowerPoint (version 2206, Microsoft Corporation, Redmond, DC, USA) and Adobe Photoshop CS6 Extended (version 13.0 ×64, Adobe Inc., San Jose, CA, USA). The content of Figure 1, Figure 2 and Figure 3 was generated by WGCNA software package. The network diagram of Figure 4a–c was generated by Cytoscape software. The ridgeline plot shown in Figure 4d was drawn using the OmicShare online tool (Guangzhou Gene Denovo Biotechnology Co., Ltd., Guangzhou, China, https://www.omicshare.com/tools/home/report/reportridgeplot.html accessed on 3 June 2022). The schematic diagram of the endoplasmic reticulum and nucleus in Figure 5 was drawn using the open-source vector graphics drawing software Inkscape (version 1.2.1, The Inkscape Project). The phylogenetic tree in Figure 6 was generated using MEGA software (version 11.0.11, Koichiro Tamura et al.) [25].

## 3. Results

### 3.1. Transcriptome Sequencing Results

The sequencing results of the 18 samples were assembled by Trinity [26], resulting in 528,309 transcripts. The original transcripts were then hierarchically clustered using Corset [27] to obtain 249,288 of the longest cluster sequence unigenes, of which 90,672 clusters were annotated into the Swiss-Prot protein database. The transcriptome analysis results of these 18 samples can be found in the paper published by Liu [20]. The subsequent analysis of this study only uses the FPKM data table and Swiss-Prot protein annotation information of these 249,288 clusters.

### 3.2. Data Input, Cleaning, and Pre-Processing

It was found that 16,734 out of 249,288 clusters had FPKM values of 0 in all 18 samples. After excluding the 16,734 rows of data, the MAD value was calculated for the remaining 232,554 rows of data. After sorting by MAD value from large to small and selecting the top 75% with MAD (>0.01), a total of 100,203 rows of data were retained to perform a cluster analysis based on Euclidean distance on the samples to obtain the clustering dendrogram shown in Figure 1a. It can be seen in the figure that the data of the *EA1* sample are abnormal and cannot be grouped with the *EA2* and *EA3* samples. Therefore, we excluded the data of the *EA1* samples and reserved the data of the remaining 17 samples for subsequent calculations.

**Figure 1 genes-13-01364-f001:**
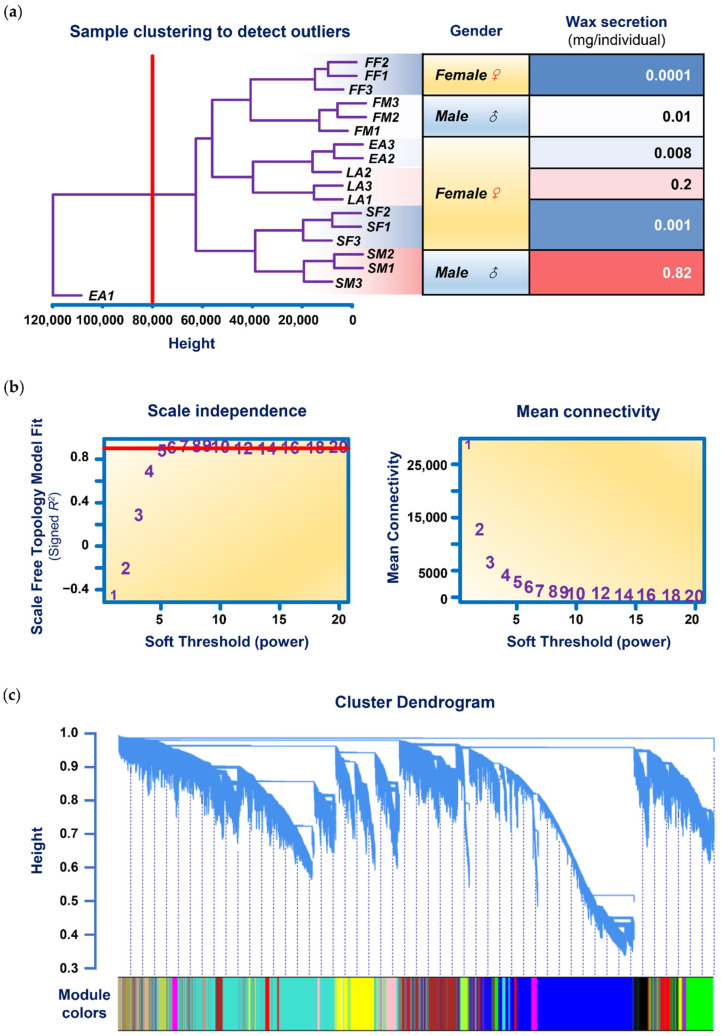
Network analysis of RNA-seq expression data in CWWS insects. (**a**) Clustering dendrogram of samples based on their Euclidean distance. *FM*, *SM*, *FF*, *SF*, *EA,* and *LA* in the figure, respectively, represent first-instar male, second-instar male, first-instar female, second-instar female, early female adult, and late female adult. The following numbers 1, 2, and 3 represent the three biological repetitions of the sample. The red vertical division line at the abscissa 80,000 in the figure indicates that the data corresponding to the abnormal sample *EA1* can be deleted by setting the parameter to 80,000 when using the *cutreeStatic()* function during data preprocessing. (**b**) Analysis of network topology for various soft-thresholding powers. The left panel shows the scale-free fit index (*y*-axis) as a function of the soft-thresholding power (*x*-axis). The number crossed by the red horizontal line (“5” in the figure) is the lowest power that the scale-free topology fitting index curve flattens when it reaches a high value. The right panel displays the mean connectivity (degree, *y*-axis) as a function of the soft-thresholding power (*x*-axis). (**c**) Clustering dendrogram of genes, with dissimilarity based on the topological overlap, together with assigned module colors.

In addition, the right part of Figure 1a also lists the wax secretion corresponding to each CWWS sample as the trait data used for the WGCNA calculation. This value is based on the wax secretion of different sample populations at the time of sample harvest to estimate the wax secretion level of a single insect individual (mg/individual).

### 3.3. Analysis of Network Topology

Constructing a weighted gene network entails the choice of the soft-thresholding power, *β*, to which the coexpression similarity is raised to calculate adjacency [28]. A scale-free topology analysis of multiple *β* was performed using the *pickSoftThreshold()* function, resulting in the results shown in Figure 1b. As can be seen from the scale independence plot on the left, the lowest *β* for which the scale-free topological fit index curve flattens out upon reaching a high value is 5 (its scale-free topology model fit is roughly 0.9). Therefore, we chose 5 as the *β* for subsequent network construction.

### 3.4. Network Construction and Module Detection

Once the network was constructed, module detection was the next step. Modules are defined as clusters of densely interconnected genes. WGCNA identifies gene modules using unsupervised clustering, i.e., without the use of prior-defined gene sets. Figure 1c shows the clustering dendrogram of all 100,203 genes. In the figure, 19 modules are distinguished based on the difference in topological overlap and are distinguished by different colors.

Gene significance (GS) was defined as the absolute value of the correlation between the gene and the trait to quantify the associations of individual genes with our trait of interest (wax secretion). For each module, we also defined a quantitative measure of module membership (MM) as the correlation of the module eigengene with the gene expression profile. This allowed us to quantify the similarity of all genes on the array to every module. Using the GS and MM measures, we could identify genes that have a high significance for wax secretion as well as a high module membership in the interesting modules and draw a color-coded table, as shown in Figure 2a (for a summary of network analysis results, see Appendix A). From this table, we observed that the blue, cyan, and tan modules have the highest association (*p* < 0.05) with wax secretion.

**Figure 2 genes-13-01364-f002:**
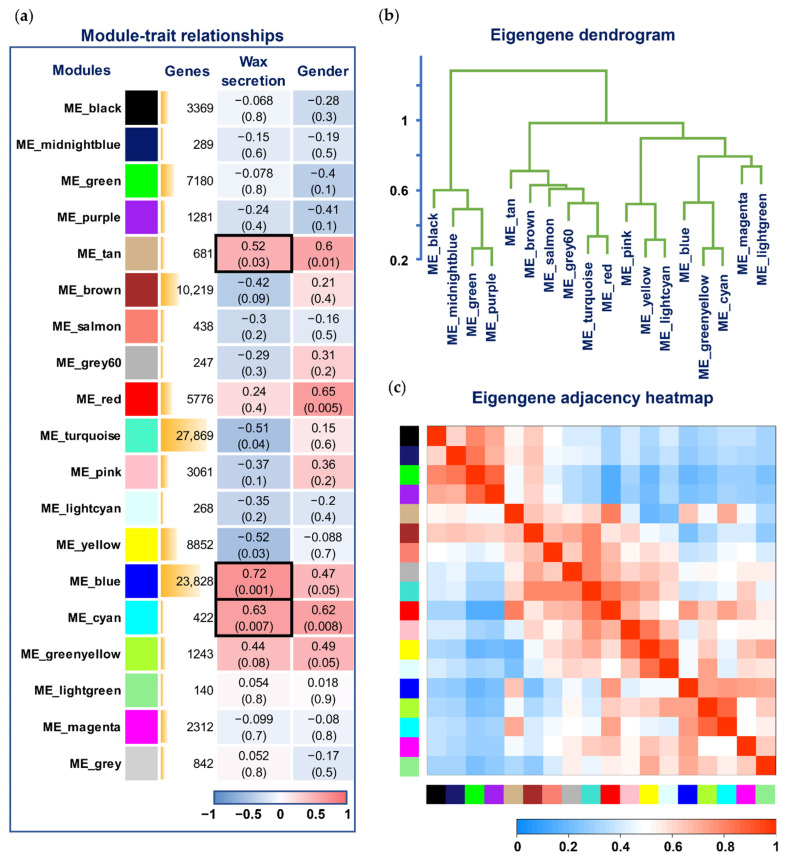
Visualization of the relationship between modules and wax secretion traits. (**a**) Module–trait associations. Each row corresponds to a module eigengene, each column corresponds to a trait, and each cell contains the corresponding correlation and *p*-value. The table is color-coded by correlation according to the color legend. The blue, cyan, and tan modules marked with black boxes have the highest association (*p* < 0.05) with wax secretion. (**b**) Hierarchical clustering dendrogram of the eigengenes in which the dissimilarity of eigengenes *E_I_* and *E_J_* is given by 1 *− cor(E_I_, E_J_)*. (**c**) The heatmap shows the eigengene adjacency *A_IJ_ =* (1 *+ cor(E_I_, E_J_*))/2.

At the same time, we also wanted to know the relationship between the modules. Figure 2b,c show the hierarchical clustering tree of modules and the adjacency relationship of each module. It can be seen from the dendrogram that the blue and cyan modules have a high correlation. Scatterplots of GS versus MM showed a highly significant correlation between *GS* and *MM* in three selected modules (Figure 3).

**Figure 3 genes-13-01364-f003:**
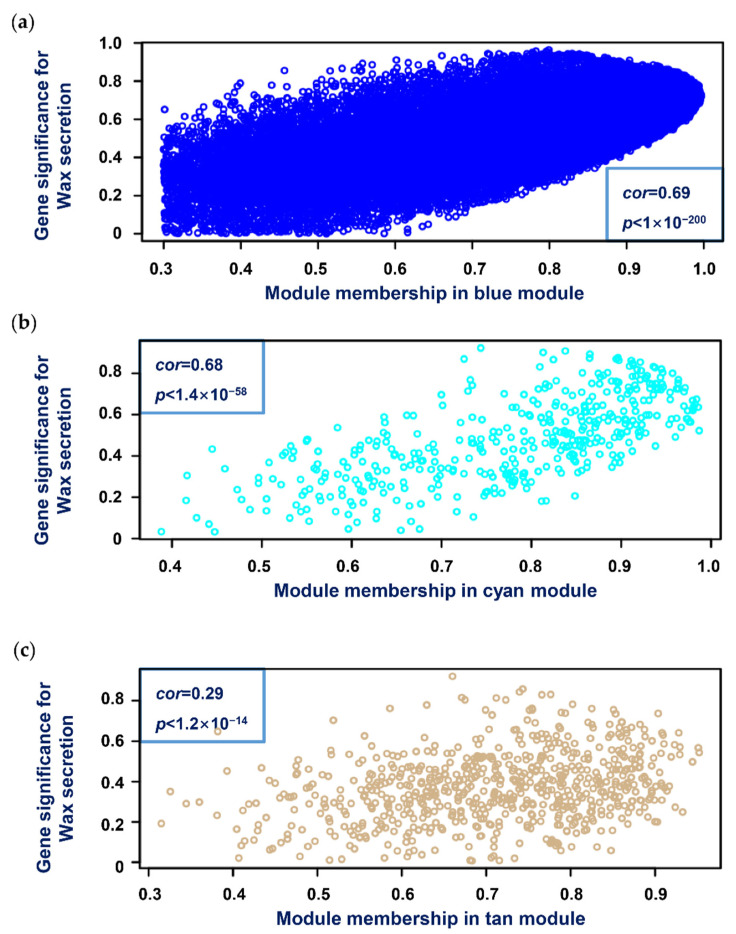
Scatterplots of gene significance (GS) vs. module membership (MM) in the blue (**a**), cyan (**b**), and tan (**c**) modules. There was a highly significant correlation between GS and MM in these modules.

### 3.5. Selection of Hub Genes

Through the *exportNetworkToCytoscape()* function in the WGCNA software package, the topological overlapping network obtained by the previous calculation was exported into a network file that can be recognized by the Cytoscape software. In this study, we screened hub genes based on the degree of each gene (called a node in the network). The degree refers to the connection between one node and another node. The larger the degree value of a node, the more connections the node has, which means that the gene that is connected with the most genes is a central gene. By using the CytoNAC weighted calculation, the top 10 genes in the three modules of blue, cyan, and tan were obtained, with a total of 30 genes (Figure 4a–c; the Cytoscape file used for the analysis can be found in Appendix A). By querying the Uniprot database [29], we obtained functional annotations of these 30 hub genes (Figure 4d).

**Figure 4 genes-13-01364-f004:**
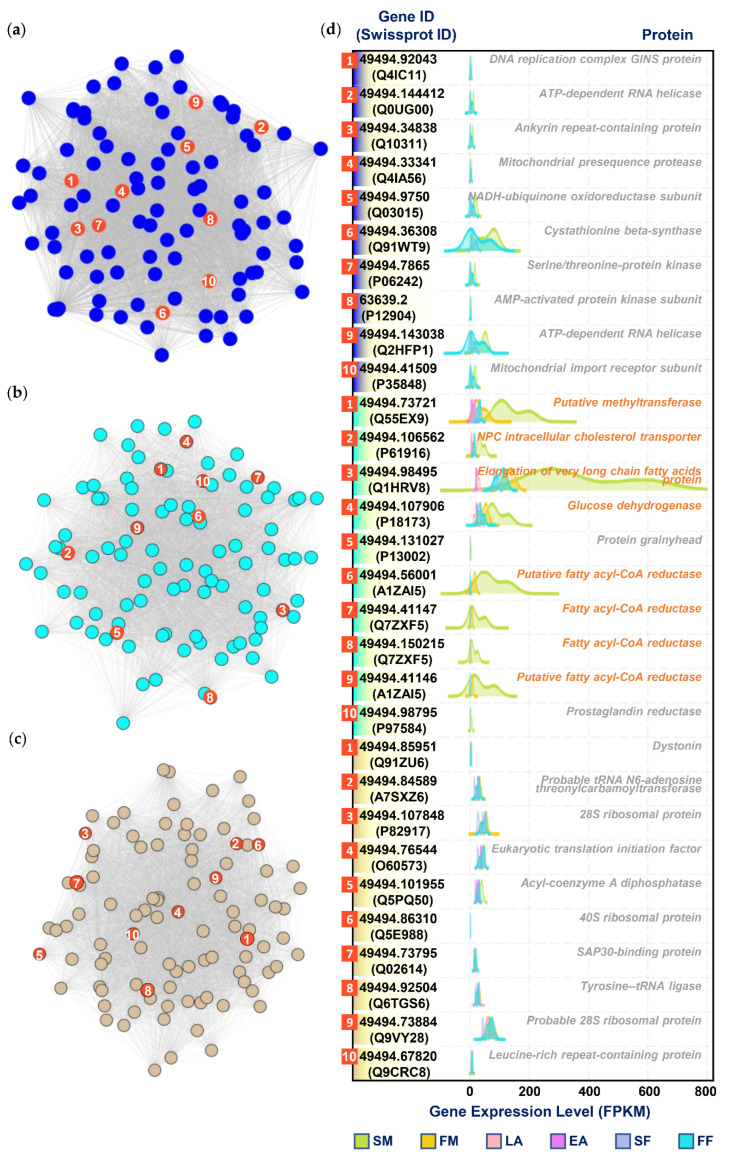
The screened hub genes related to wax secretion are displayed through a visual network structure diagram and ridgeline plots. (**a**–**c**) represent the visualization of the network connections between the top 100 genes with the most connections in the blue (**a**), cyan (**b**), and tan (**c**) modules, which were generated by the Cytoscape software. The nodes marked in red are the top 10 nodes (genes) with the largest degree calculated by the CytoNAC plugin, and the 1 to 10 numbers marked in the nodes are the order of the degree of the nodes in descending order. (**d**) Ridgeline plots of the expression levels of the selected hub genes in each sample. *FM*, *SM*, *FF*, *SF*, *EA,* and *LA* in the figure, respectively, represent first-instar male, second-instar male, first-instar female, second-instar female, early female adult, and late female adult. On the left side of the panel is the ID corresponding to the gene in the sequencing result, the annotated Swissprot ID is in brackets, and the text on the right side of the panel corresponds to the annotated protein function. The genes marked with orange are the genes whose expression levels in *SM* are significantly higher than those of the other samples.

From the ridgeline plots of the expression levels of hub genes in each sample shown in Figure 4d, it can be found that there are eight genes (marked with orange in Figure 4d) whose expression levels in second-instar males (*SM*) are significantly higher than those in the other samples. Among them, ELOVL and FAR are important catalytic enzymes in fatty acid metabolism, which are located in the endoplasmic reticulum membrane and the peroxisome membrane in cells, respectively (Figure 5). Functionally, ELOVL could be implicated in the synthesis of very-long-chain fatty acids. FAR catalyzes the reduction of saturated and unsaturated C_16_ or C_18_ fatty acyl-CoA to fatty alcohols. In parallel, it is also required for wax monoester production since fatty alcohols also constitute a substrate for their synthesis.

**Figure 5 genes-13-01364-f005:**
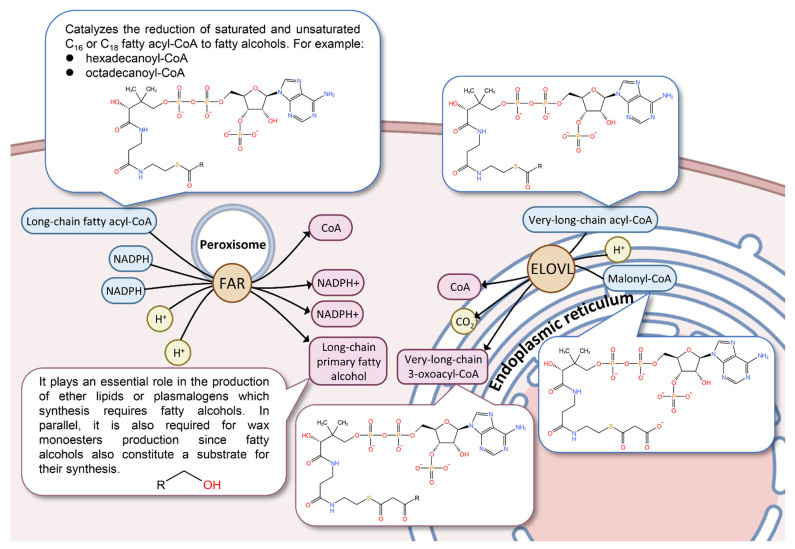
Schematic diagram of intracellular long-chain primary fatty alcohol and very-long-chain 3-oxoacyl-CoA synthesis pathway. This schematic diagram is based on the functional and subcellular location introduction of elongation of ELOVL (UniprotID: Q1HRV8) and FAR (UniprotID: Q7ZXF5) in the Uniprot database. The text, chemical structure, and abbreviation in the figure are all from the Uniprot database. Both the peroxisome and the endoplasmic reticulum are schematic, and their size and scale are not drawn according to the actual size and scale of the cells.

## 4. Discussion

From the results, we believe that the expected result can be obtained by the WGCNA calculation method, that is, the hub genes related to the wax secretion of CWWS insects. In the routine RNA-seq analysis process, we usually presume that there are related genes that we need to pay special attention to through the published articles and the specific metabolic pathways that have been reported and then verify whether the characteristics of the genes we are concerned about meet our expectations or find some new interesting genes according to analysis methods such as the differential expression of genes and GO and KEGG enrichment. The advantage of using WGCNA analysis is that the whole process of screening hub genes from the beginning to the end is completed by calculation, and a large amount of data are summarized and analyzed, which reduces the process of manual screening. We believe that this method is particularly suitable for analyzing and screening unknown genes involved in the regulation of specific traits.

Before proceeding with module construction, we first performed a hierarchical clustering analysis on the samples. According to the description of the R language function *dist()* [30], the distance between samples was calculated using the Euclidean distance method:(1)$$distance=\sqrt{\overset{k}{\underset{i=1}{\sum }}{\left(x_{i}-y_{i}\right)}^{2}}$$

In Equation (1), *x_i_* and *y_i_* refer to the FPKM value of the *i* gene of sample *x* and *y*, and *k* refers to all genes [31]. When using the *hclust()* function for hierarchical clustering [32], we chose unweighted average linkage clustering (UPGMA) as the linkage criteria for clustering. In this study, the expression levels of many genes in the *EA1* samples were significantly different from the other samples. For example, judging from the values listed in Appendix A, the FPKM value of Cluster-49494.88729 in other samples can reach a minimum of 2.08, but in the *EA1* sample it is only 0.52, and there are many similar genes. This resulted in the inability of the *EA1* sample to cluster with the other samples. The reason for this situation, we speculate, may be a problem of sample contamination or insufficient freshness of the *EA1* sample at the time of collection or a degradation or pollution problem during the extraction of total RNA. However, whatever the case, the data from the *EA1* sample should not be used for subsequent calculations.

How to screen hub genes is a key problem in this study. In screening technology, analysis based on a centrality measurement has been proven to be an effective means to identify essential proteins [24]. Centrality is a concept used in network analysis to measure the degree to which a node in the network is close to the center of the whole network. This degree is expressed in numbers, which is called centrality. Its main measurement indicators are degree centrality (DC), closeness centrality (CC), between centrality (BC), and eigenvector centrality (EC) [33]. The DC measures the degree to which a node in the network is connected to all other nodes [34]:(2)$$C_{D}\left(u\right)=\left|N_{u}\right|$$

In Equation (2), |*N_u_*| is the number of node *u*’s neighbors. In this study, we also considered the weight of edges [35]:(3)$$C_{D,\;W}\left(u\right)=\sum_{v\in N_{u}}W\left(u,\;v\right)$$

In Equation (3), *N_u_* is the node set containing all the neighbors of node *u,* and *W*(*u*, *v*) is the weight of the edge connecting node *u* and node *v*. In this study, we used the weighted DC index to sort the nodes in the network and screen out the hub genes. CC reflects the closeness between a node and other nodes in the network [36]. Unlike degree centrality, which considers direct connections, CC considers the average length of the shortest paths from each node to other nodes. That is, for a node, the closer it is to other nodes, the higher its centrality. In this study, the distance between two nodes in the network is not the result of the WGCNA calculation. It is the result of the optimal placement of nodes obtained by Cytoscape software according to the calculation. Therefore, we believe that the CC index does not apply to this study. BC is used to measure the number of times a vertex appears in the shortest path between any two other vertex pairs, thereby characterizing the importance of nodes [37]. The BC index is the attribute of studying the nodes connecting two or more networks, and it does not apply to the node screening within the network in this study. We used the CytoNCA plugin to calculate that the BC value of each node is 0, which also proves this. EC is a measure of the influence of a node on the network [38]. A node has high EC if it is pointed to by many nodes (which also have high EC). We also used the CytoNCA plugin to calculate the weighted EC indices of the three modules, which are listed in Table 1. We found that the results are very interesting. The top 10 genes selected in each module are sorted from large to small according to DC and EC. Except for the fact that some genes are sorted in a different order, the gene compositions in the screening results of DC and EC in the module are the same. Such results also validate that our method for screening hub genes is efficient and verifiable.

When conducting this experiment, we found the limitations of the application of WGCNA in this study, namely, the insufficient sample size and the amount of phenotypic data. The main reason is that the CWWS insects used in the experiment have limited combinations of different sexes and insect states, so only six phenotype data points were set in this experiment. However, we also refer to the description of the sample size by Peter Langfelder and Steve Horvath [39], authors of the WGCNA software package, and believe that the sample size for WGCNA analysis should be no less than 15, and if possible, there should be at least 20 samples. Considering that we obtained transcriptome data for 18 samples, the minimum sample size required for WGCNA analysis was reached, so we tried to analyze these samples using the WGCNA method, and the results proved that this method is suitable for this study. Even so, we suggest that similar studies should be performed with larger samples. Larger samples usually lead to more robust and refined results.

Very-long-chain fatty acids are the precursors of sphingolipids and glycerolipids, which are essential components of the cell membrane structure and participate in various cellular biological processes [40]. In addition, fat can store and provide energy and is an important energy supply material for insects. The proportion of unsaturated fatty acids and the fatty acid content in insects also affect the cold tolerance of insects [41]. According to the composition of long hydrocarbon chains ending in carboxyl groups, fatty acids are divided into short-chain fatty acids (containing five or fewer carbon atoms), medium-chain fatty acids (containing six to twelve carbon atoms), long-chain fatty acids (containing more than twelve carbon atoms), and very-long-chain fatty acids (containing twenty-two or more carbon atoms) [42]. Fatty acid synthesis mainly occurs in the endoplasmic reticulum. Each cycle extends the initial acetyl-CoA by two carbon atoms, which can be repeated up to seven times to form palmitic acid (C16:0). Further growth requires elongases for long-chain or very-long-chain fatty acids (ELOVLs), which can extend fatty acids greater than 14 carbon atoms through a fatty-acid condensation reaction [43]. ELOVLs have been isolated from a variety of organisms, including yeast, mammals, plants, and other species. Although ELOVLs are relatively well-characterized in vertebrates, little is known about these enzymes in insects. At present, the ELOVLs of 12 species from Arthropoda Insecta have been reported in GenBank. We translated the *ELOVL-like* genes annotated in the CWWS insect transcriptome into protein sequences, compared them with the ELOVL protein sequences from humans and 12 insect sources, and established a phylogenetic tree (Figure 6). It was found that the ELOVL of *E. pela* can be grouped with *Drosophila melanogaster* and *Tenebrio molitor*. The ELOVL protein of CWWS insects has not been reported. Given its important function in the process of lipid metabolism, it is necessary to conduct in-depth research on it.

**Figure 6 genes-13-01364-f006:**
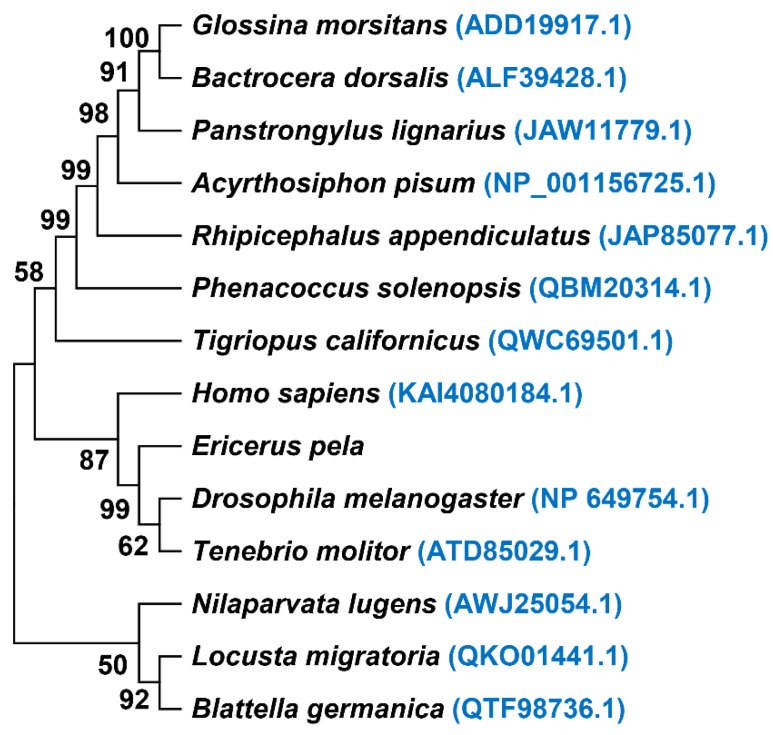
Phylogenetic tree of elongation of very-long-chain fatty acid protein (ELOVL) in 14 species. The phylogenetic tree was constructed using the neighbor-joining method with 1000 bootstrap replicates. Each species was followed by its GenBank accession number.

The *FAR* gene of *E. pela* is the first reported gene related to wax secretion in CWWS insects [16]. It has a sequence of 520 amino acids, is expressed in the male testis, digestive tract, body fat, and Malpighian tubules, and was localized by immunofluorescence only in the wax glands and testis [17]. We compared the four hub genes (Cluster-49494.56001, Cluster-49494.41147, Cluster-49494.150215, and Cluster-49494.41146) obtained in this study, annotated as *FARs*, with the *FAR* of *E. pela* reported by Yang (GenBankID: AGK27745) for protein. The BLAST alignment showed that three sequences could be aligned to different positions of the AGK27745 sequence (Figure 7), which also verifies that our research is consistent with the results reported by Yang.

**Figure 7 genes-13-01364-f007:**
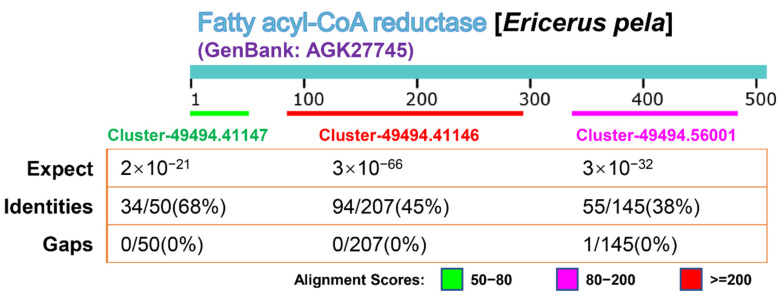
The results of the BLAST alignment of the three clusters annotated as *FAR* genes in the CWWS insect transcriptome and the reported FAR protein sequences in GenBank. The three clusters were aligned to three different positions in the FAR protein sequence (GenBankID: AGK27745). Comparison method: compositional matrix adjustment. Expect (E-value) indicates the possibility of random matching. The larger the E-value, the greater the possibility of random matching. When the E-value is close to zero or zero, it is a perfect match. Identities: The percentage of matched bases in the total sequence length. Gaps: Insertion or deletion.

In addition, we also found that the expression levels of three genes, *methyltransferase*, *glucose dehydrogenase* (FAD, quinone), and *NPC intracellular cholesterol transporter*, were significantly increased in second-instar males, and were screened as hub genes. *Glucose dehydrogenase* (FAD, quinone) is essential for epidermal modification during *Drosophila melanogaster* development. Its function is to catalyze quinone and D-glucose to quinol and D-glucono-1,5-lactone (Figure 8). The cofactor is FAD. *Glucose dehydrogenase* is widely expressed and secreted in the seminiferous ducts of *D. melanogaster*. This highly conserved expression pattern suggests that this enzyme plays an important role in female fertility [44]. In CWWS insects, the regulatory functions of these genes and whether they are related to wax ester metabolism have not yet been revealed, which is worthy of further study.

**Figure 8 genes-13-01364-f008:**
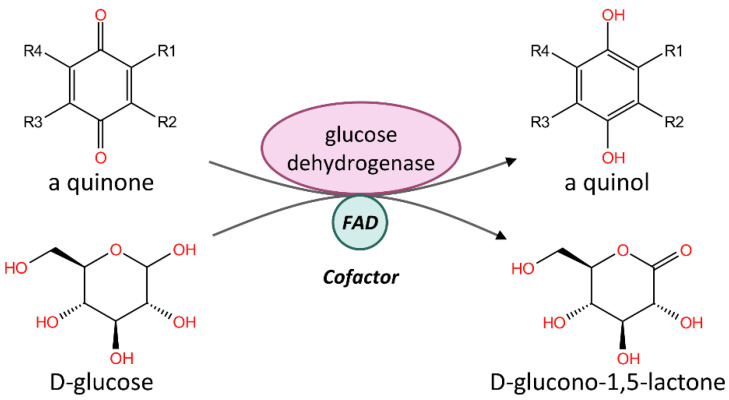
Glucose dehydrogenase with FAD as a cofactor catalyzes quinone and D-glucose to quinol and D-glucono-1,5-lactone in *Drosophila melanogaster*. This schematic diagram is based on the function of *Glucose dehydrogenase* (FAD, quinone) (UniprotID: P18173) in the Uniprot database.

## 5. Conclusions

In summary, we used the WGCNA to analyze the transcriptome of the CWWS and screened out three modules as highly significant for wax secretion with the 30 hub genes related to wax biosynthesis. This indicates that the WGCNA method could be used widely in target gene screening of the insect genome and transcriptome with high effectiveness.

## Figures and Tables

**Table 1 genes-13-01364-t001:** Calculating the degree centrality (DC) and eigenvector centrality (EC) indices of each gene in the three modules using CytoNCA.

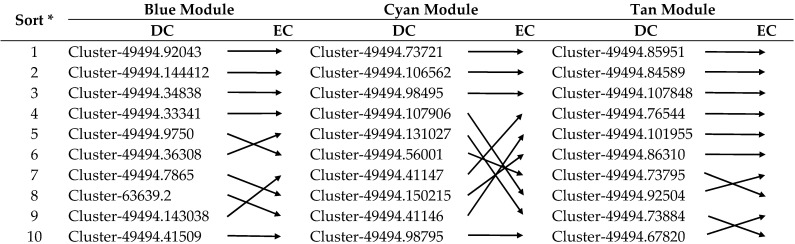

* All genes in each module were sorted from large to small according to DC and EC indices. The top 10 genes are listed in the table. The gene ID is omitted in the EC column, and the arrows are used to visually display the ranking differences of the genes in the DC and EC columns. The horizontal arrows indicate that the same gene is ranked in the same position in the two columns, the arrows pointing to the lower right indicate the genes in the EC column have lower ranks, and the arrows pointing to the upper right indicate the genes in EC column have higher ranks.

## Data Availability

All data and results used in this study were provided to readers as Appendix A.

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
