# Peer review of "Identification of the Key Pathways and Genes Involved in the Wax Biosynthesis of the Chinese White Wax Scale Insect (*Ericerus pela* Chavannes) by Integrated Weighted Gene Coexpression Network Analysis"

_genes, 2022, doi:10.3390/genes13081364_

Round 1

Reviewer 1 Report

1. Extensive English editing is required, mostly from the methodology, present and future tenses were used.

2. The objective for this study is not clear...

3. Why use the WGCNA?

4. Line 38: add against after defending...

5. Line 91: reconstruct as,......the wax ester regulatory pathway of the CWWS insect is still not clear

6. Please correct the English in all the text

7. The discussion should have focused more on comparing the different methodology in identifying the hub gene or genes of interest. 

In general, it is not clear how the WGCNA method is easier

Author Response

Many thanks to the reviewer for the comments and suggestions on our manuscript. In response to your comments and suggestions, we will reply as follows:

Point 1: Extensive English editing is required, mostly from the methodology, present and future tenses were used.
Point 4: Line 38: add against after defending...
Point 5: Line 91: reconstruct as,......the wax ester regulatory pathway of the CWWS insect is still not clear
Point 6: Please correct the English in all the text

Response: The revised manuscript has been proofread by a professional English editing service company.

Point 2: The objective for this study is not clear...

Response 2: The goal of this research is to analyze 18 groups of the Chinese white wax scale data from different genders and insect states by the WGCNA method, and finally obtain the hub genes related to the wax biosynthesis.

Point 3: Why use the WGCNA?

Response 3: With the continuous advancement of sequencing technology and the gradual reduction in sequencing costs, more and more researchers have begun to design multi-sample RNA-seq studies. Multiple samples can analyze gene expression changes under multiple conditions, making scientific research data more substantial, systematic and convincing. However, at the same time, multi-samples bring a large amount of grouped data, and the workload required for the traditional difference analysis between two groups is very large, which is not conducive to efficient research. Weighted gene co-expression network analysis (WGCNA) can summarize and organize complex data and efficiently study the overall expression rules of genes. At the same time, it can systematically feed back the interaction pattern between genes in the samples, help to mine key genes, predict gene function, and achieve the purpose of significantly improving gene screening. Therefore, WGCNA plays an important role in analyzing multi-sample RNA-seq data.
        As we used 18 sets of transcriptome data in this study, and there were differences in the phenotypic data of wax secretion among each group, we believed that these factors met the conditions for WGCNA analysis, so the WGCNA analysis method was used in this study.

Point 7: The discussion should have focused more on comparing the different methodology in identifying the hub gene or genes of interest.

Response 7: As suggested by the reviewers, the method used to screen for hub genes is indeed an important focus of this study. We accept the reviewer's suggestion and elaborate in the third paragraph of the "4. Discussion" section why we chose the degree centrality index as the basis for screening hub genes in this study, and compared it with other centrality indices.

Point 8: In general, it is not clear how the WGCNA method is easier

Response 8: The WGCNA analysis method is not "easier" in actual operation, and this process requires the server to perform a long calculation. What we want to express in the manuscript is that the analysis process of WGCNA, from the beginning to the final acquisition of hub genes, is performed computationally with little manual intervention, which we think is a major feature of WGCNA.

Reviewer 2 Report

The authors identified key genes and pathways involved in the wax biosynthesis of the Chinese white was scale using the network analysis. This work gives a pleasing impression of thoughtfulness. However, there are a few factors that would improve the clarity of the paper, listed below.

  • Line #58: cite the paper related to “The study of wax ester biosynthesis was first carried out in Arabidopsis thaliana”.

  • Line #177: Can you explain why EA1 was clustered far away from the rest of the samples?

  • Can you provide results of the module preservation and connectivity as the supplementary data?

  • Please discuss the limitations of this study, i.e. small sample size.

  • Writing should be improved.

Author Response

First of all, many thanks to the reviewers for acknowledging our work, we are deeply inspired by your evaluation of our work. In response to your comments and suggestions, we will reply as follows:

Point 1: Line #58: cite the paper related to “The study of wax ester biosynthesis was first carried out in Arabidopsis thaliana”.

Response 1: In the revised manuscript, we have added corresponding reference citations.

Point 2: Line #177: Can you explain why EA1 was clustered far away from the rest of the samples?

Response 2: In this study, the expression levels of many genes in EA1 samples were significantly different from other samples. For example, judging from the values listed in Supplementary Material S1, the FPKM value of Cluster-49494.88729 in other samples can reach a minimum of 2.08, but in the EA1 sample, it is only 0.52, and there are many similar genes. This resulted in the inability of the EA1 sample to cluster with other samples. The reason for this situation, we speculate, may be the problem of sample contamination or insufficient freshness of the EA1 sample at the time of collection or the degradation or pollution problem during the extraction of total RNA. However, whatever the case, the data from the EA1 sample should not be used for subsequent calculations. We have added an explanation of this part to the second paragraph of the manuscript "4. Discussion".

Point 3: Can you provide results of the module preservation and connectivity as the supplementary data?

Response 3: Of course. We have added the Cytoscape software format network of the three modules analyzed in this study to the uploaded revised manuscript as Supplementary Material S6.

Point 4: Please discuss the limitations of this study, i.e. small sample size.

Response 4: We accept your suggestion and have added an analysis of the limitations of this study in the fourth paragraph of the "4. Discussion" section.

Point 5: Writing should be improved.

Response 5: The revised manuscript has been proofread by a professional English editing service company.